# 2D CNN-Based Multi-Output Diagnosis for Compound Bearing Faults under Variable Rotational Speeds

**Minh-Tuan Pham** [1], **Jong-Myon Kim** [2] and **Cheol-Hong Kim** [1,*]

1    School of Computer Science and Engineering, Soongsil University, Seoul 06978, Korea; dlwpak4@soongsil.ac.kr
2    School of IT Convergence, University of Ulsan, Ulsan 44610, Korea; jmkim07@ulsan.ac.kr
*    Correspondence: cheolhong@ssu.ac.kr; Tel.: +82-2-820-0674

**Abstract:** Bearings prevent damage caused by frictional forces between parts supporting the rotation and they keep rotating shafts in their correct position. However, the continuity of work under harsh conditions leads to inevitable bearing failure. Thus, methods for bearing fault diagnosis (FD) that can predict and categorize fault type, as well as the level of degradation, are increasingly necessary for factories. Owing to the advent of deep neural networks, especially convolutional neural networks (CNNs), intelligent FD methods have achieved significantly higher performance in terms of accuracy. However, in addition to accuracy, the efficiency issue still needs to be weathered in complicated diagnosis scenarios to adapt to real industrial environments. Here, we introduce a method based on multi-output classification, which utilizes the correlated features extracted for bearing compound fault type classification and crack-size classification to serve both aims. Additionally, the synergy of a time–frequency signal processing method and the proposed two-dimensional CNN helped the method perform well under the condition of variable rotational speeds. Monitoring signals of acoustic emission also had advantages for incipient FD. The experimental results indicated that utilizing correlated features in multi-output classification improved both the accuracy and efficiency of multi-task diagnosis compared to conventional CNN-based multiclass classification.

**Keywords:** convolutional neural network; multi-output classification; acoustic emission; time–frequency domain; bearing fault diagnosis





## 1. Introduction

Electric machines have been widely used and play an undeniable role in industrial applications, as well as in machinery serving life. Continuous operation under various conditions (temperature change, overload, high moisture level, etc.) causes inevitable faults for machines and exerts adverse effects on safety standards, production quality in factories, cost, and downtime. Based on some surveys of the IEEE Industry Application Society and other related organizations, bearings account for approximately 40% of machine fault causes [1]. This has caused alarm and heightened the need to develop bearing fault diagnosis (FD) methods that will prevent unwanted incidents and ensure the reliability and safety of sophisticated systems.

Nowadays, industrial companies increasingly seem to find FD an essential task to keep track of desirable performance during the production processes. The competitiveness of a company is strongly related to its level of intelligence in the FD process. Therefore, most industrial companies desire to improve their performance by enhancing their capability to handle faults. The common process of FD is summarized into two stages: (1) observing the behavior of monitoring objects; and (2) determining the existing faults and their nature, explaining the root causes. Thus, the level of a smart factory depends on its ability to utilize information from the data.

Some experimental and theoretical studies have been conducted in the field of FD, especially regarding bearings. In general, the diagnosis issues are solved by modeling the

physical model bearing or finding the relationship between bearing defects and the corresponding characteristic of monitoring signals that carry the bearing health information. Various modalities have been utilized for monitoring bearings—such as vibration [2–5], stator current [6,7], thermal imaging [8], electromagnetic signals [9], and acoustic emission (AE) [10–13]. In general, these methods are categorized into knowledge-based, physical model-based, and data-driven approaches, with the help of signal processing analysis in the time-, frequency-, or time–frequency domains. However, attaining diagnostic accuracy is more challenging in reality, as incipient fault detection with a small signal-to-noise value is required. The presence of other complicated working conditions, such as variable rotational speeds and the requirement of predicting degradation levels, also make manual modeling effort-intensive. Among these approaches, data-driven methods have the advantage of reducing effort in modeling or accumulating prior knowledge of interests. With the advent of machine learning (ML), data-driven approaches have significantly impacted FD processes owing to their automation, simplicity, and effectiveness. This opens a new era in the field—namely, intelligent FD—which is based on ML algorithms, including principal component analysis (PCA), artificial neural network (ANN), support vector machine (SVM), and k-nearest neighbors (k-NN), to learn from the acquired data and adapt the acquired knowledge to predict the presence of bearing faults with high accuracy [14,15]. Furthermore, deep learning (DL) has achieved even better performance in noisy environments and versatile operation condition constraints [16–18].

Besides the accuracy in diagnosis, the real-time capability of FD is also a critical aspect that needs to be considered, especially in hazardous industrial types of machinery. Efficiency is also essential for reducing the computational resources of the entire monitoring system, especially in limited-resource devices (e.g., handheld devices). Although DL-based methods, especially CNN, show high performance, latency is their critical problem that needs to be considered because they consume a large number of computing resources. Some studies have focused on reducing the number of MAC (multiply–accumulate operations) and the number of parameters, thus indirectly reducing the latency of diagnosis inference. For example, simple CNN architectures were designed [19,20]. The scenario of fault type classification works well on limited-resource systems (e.g., embedded systems) by CNN-based methods using a small input image size or model established by a neural architecture search [21,22]. However, the efficiency of more complex diagnosis scenarios still needs to be considered to fulfill the diagnosis process, which could play a role as an intermediate stage for fault prognosis to estimate the longevity of the bearing. M.T. Pham et al. [10] proposed a method based on the efficient-net (CNN) to predict not only bearing fault types but also the level of degradation, but the proposed multiclass classification shapes a network with many classes. This method requires a large amount of computational resources. In an effort to utilize correlated features in terms of multiple classification tasks, Shen et al. [23] proposed a multiple-label framework applied to raw vibration signals to predict single faults with the level of degradation. The simplicity of signal representation made it difficult to ensure stability under conditions of variable rotational speeds and noisy working environments.

This paper proposes a method based on CNN multiple-output classification to address the diagnosis problem in complicated scenarios. The proposed method initially utilizes time–frequency analysis applied to AE signals to represent fault features under conditions of varying rotational speeds owing to the nonstationary property of monitoring signals. The proposed CNN multiple-output model, which utilizes the synergy of correlated features to solve two tasks of fault type diagnosis and crack size diagnosis (level of degradation), will be applied if there are existing faults detected by a preliminary anomaly detection model (ADM). This multi-task learning process can improve the overall efficiency and accuracy. The proposed method can also help to utilize a part of the training samples lacking labels.

The rest of this paper is organized as follows: Section 2 reviews works related to intelligent bearing FD. Section 3 details the proposed method based on the multiple-output CNN model applied for two-dimensional (2D) spectrogram representation. Section 4

conducts experiment to evaluate the proposed diagnosis method, accompanied by results, explanations, and comparisons. Finally, Section 5 presents our conclusions.

## 2. Related Works

Over the past decade, the advent of ML has created great tools to automatically generalize and learn from data. Hence, intelligent bearing FD is derived from a combination of conventional signal processing methods and ML tools. On the one hand, signal processing methods can be performed in single domains (i.e., time-domain, frequency-domain) [24,25] or in the time–frequency domain [26,27]. In contrast, there have been numerous ML methods applied for bearing FD. First, the k-NN classifier is one of the simplest and most basic tools utilizing the shortest distance in the feature space among training samples, which does not consider the data distribution [28]. Among the measurement metrics, the Euclidean distance among samples is the most popular in k-NN [29]. In addition to the advantage of simplicity in implementation, k-NN shows its disadvantage in computational expenses depending on the number of data dimensions. In the field of bearing FD, Pandya et al. [30] proposed APF-kNN applied to AE signals that are preprocessed by the Hilbert–Huang transform. Baraldi et al. [31] integrated k-NN and binary differential evolution to diagnose bearing states under varying conditions. Second, ANN also contributes to the diagnosis area, which is based on the principles of the human brain. An ANN is composed of input, hidden, and output layers, which are connected consecutively to learn the knowledge from the historical data [32]. Owing to its classification capability, Yu et al. [33] proposed a method for bearing condition diagnosis using input data from energy features created by EMD. Moreover, Ben Ali et al. [34] proposed an ANN-based method using input data from the energy entropy calculated by IMFs. The disadvantage of ANNs is the requirement of a large amount of training data and the difficulty of network scale selection. Thus, it needs a lot of effort when applying it in reality. Next, the limitation of ANN is weathered by the support of SVM, which is able to separate the group of data by maximizing the gap between classes in the search space [35]. For example, Jiang et al. [36] applied SVM for multi-source monitoring signals represented in the time domain, which were acquired from some sensors to diagnose fault sources (gear, bearing, and rotor). In an effort to mitigate the downside of SVM when the number of votes for classes is equivalent (decision conflicts), Hui et al. [37] proposed a method using an improved version of SVN, namely SVM-Dempster Shafer, to detect the anomaly accurately.

Recently, the movement toward DL has helped bearing FD perform better while reducing the manual processes compared to conventional FD methods. DL, especially CNN-based methods, can solve more complicated diagnosis problems by the capability of fault feature extraction. CNN is initially applied directly for acquired monitoring signals in the time domain; for example, Bhadane et al. [38] used statistical parameters as input features to feed the CNN. Other methods also utilize one-dimensional (1D) CNN models applied to raw monitoring signals in terms of simplicity and efficiency. For instance, Shao et al. [39] proposed a hybrid model that combined 1D CNN and SVM with the support of an improved swarm optimization algorithm to enhance the performance and convergence speed. Zhou et al. [40] proposed an 1D CNN-based fusing frequency feature matching algorithm to extract key frequency features in the signal spectrum for bearing fault diagnosis under noisy environments. Ince et al. proposed an efficient 1D CNN-based method that adapts an inherent adaptive design to combine a feature extractor and classifier into a single learning body, with the input data being raw signals [41]. Zan et al. built a multi-dimension input CNN model based on 1D CNN which utilizes multiple input layers rather than one [42]. Moreover, 2D CNNs are increasingly used owing to their power in image processing. For example, Li et al. proposed a 2D CNN-based method applied to wavelet packet transform images [43]. Zhuang et al. proposed a method utilizing stacked residual dilated convolutions with high denoising capability; however, it has high complexity in terms of the number of parameters and training time [44]. Ma et al. proposed a method based on the idea of unsupervised learning with the loss function to solve the

problem of diagnosis under variable rotational speeds [45]. Haedong et al. [46] proposed a CNN-based method using orbit plot images as input data to classify the fault modes [46]. Yuan et al. [27] proposed a combination of CNN and SVM to construct a network framework for bearing FD. In reality, other aspects are also considered, for example, Han et al. [47] proposed an adversarial learning framework for alleviating the overfitting problem due to the lack of labeled data. The efficiency of CNN-based methods has received attention in the field of bearing FD [21,22]; however, they still need to be considered in complicated diagnosis conditions.

## 3. Proposed Method

In this section, we introduce our proposed method based on 2D CNN multi-output classification for both the aims of compound fault type diagnosis and level of degradation diagnosis. After providing an overview of the proposed method, we clarify the characteristics of the signal processing technique used, and the terms related to multi-output classification in FD.

CNNs are powerful tools for dealing with the issue of feature extraction, especially with 2D image data. Therefore, the CNN model was compatible with the 2D representation of the signal spectrogram in the time–frequency domain, which could effectively represent fault features for acquired nonstationary signals under variable rotational speed conditions. Neural networks based on CNNs consist of three layers: convolutional, pooling, and fully connected layers. Whereas convolutional and pooling layers have been used for establishing feature extractors, fully connected layers are often used to construct classifiers. As Figure 1 shows, the proposed method contained two sub-models constructed based on basic layers: the ADM and the multi-output FD model. On the one hand, the first model served for binary classification between normal and abnormal states, which were pre-screened based on a practical condition (bearings usually were in a normal state). In contrast, the second model was activated whenever an anomaly was detected to perform further tasks of fault type diagnosis and degradation level diagnosis.

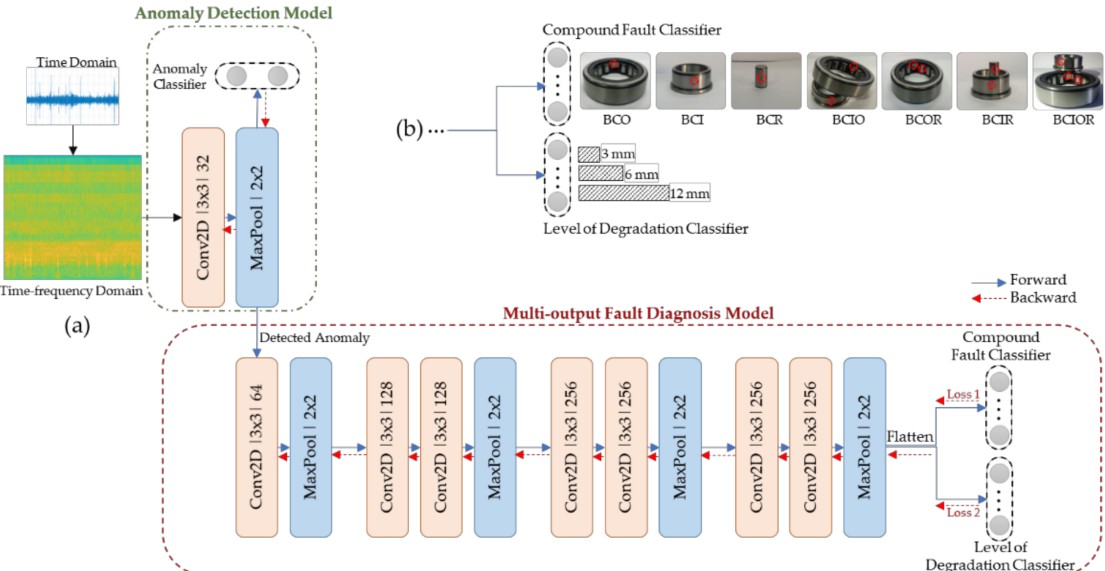

**Figure 1.** (**a**) Proposed CNN-based multi-output classification framework (Conv2D | x | y means 2D CNN | kernel size | channels; Maxpool means Max pooling operation); (**b**) Illustration of two classifiers for two diagnosis aims: compound fault types: Outer raceway (BCO), Inner raceway (BCI), Roller (BCR), Inner and outer raceway (BCIO), Outer raceway and roller (BCOR), Inner raceway and roller (BCIR), Inner and outer raceway and roller (BCIOR); Crack size: 3, 6, 12 mm.

### 3.1. Time–Frequency Analysis for AE Signals

In the working condition of variable rotational speed, single domains (time domain or frequency domain) could not adequately represent the fault features owing to the nonstationary characteristic of monitoring signals [22,48,49]. In contrast, time–frequency utilized the synergy of both domains to analyze the signal spectrum of the transient signal. Among time–frequency analysis methods, wavelet packet transform (WPT) has been widely used owing to its flexibility in changing the resolution according to the frequency range. WPT variants are related to the Gabor transform being effective when using a low bandwidth duration and transient signals. This study used the Constant-Q Transform (CQT), a variant of WPT, to create spectrogram images from acquired monitoring signals. CQT has the advantage of representing a signal at low frequencies and solves the problem of mapping frequency on a logarithmic scale. Low-frequency components in the acquired AE signals contain more meaningful bearing information than those in the high-frequency range because of high-frequency noise. CQT needs to consider the following factors to establish: (1) window $g_k$, which was real-valued and even functions—in the frequency domain, the Fourier transform of $g_k$ was defined in the interval $[-F_s/2, F_s/2]$; (2) the sampling rate $\omega_s$; (3) the number of bins per octave, b; and (4) the minimum and maximum frequencies, $\omega_{min}$ and $\omega_{max}$, respectively.

After being converted into spectrograms, all samples were removed from their redundant borders and rescaled into the size of 224 × 224 by linear interpolation to ensure the ability to represent features for both fault types and levels of degradation. Figure 2 illustrates single-fault signal spectrograms at various levels of degradation under the condition of variable rotational speeds.

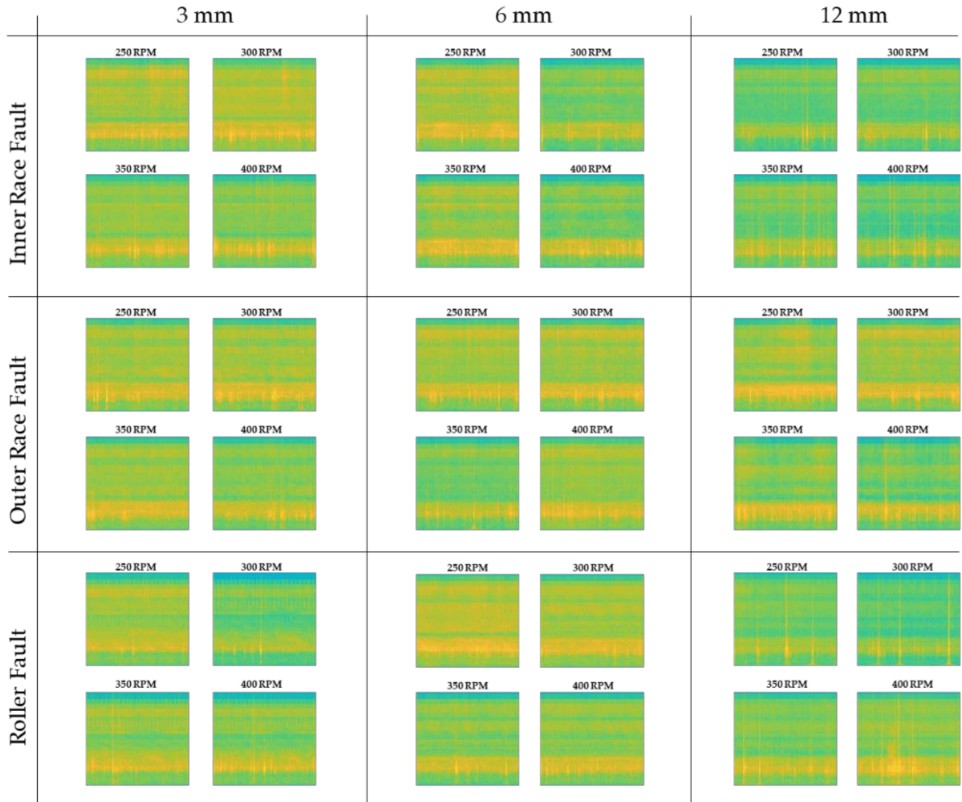

**Figure 2.** 2D spectrograms of single fault types with various levels of degradation at different rotational speeds.

### 3.2. Anomaly Detection

Owing to the simplicity of the task, the ADM was constructed using three layers. First, the convolutional layer convolved across the input image size to extract a high-abstract feature map. Second, a max-pooling layer was used to downsample the dimensions of the feature maps. The pooling operation acted as a feature summarization to improve the robustness of the model with the change of feature location in the input image data. Third, both the number of parameters and computational resources were reduced by using pooling operations. The last layer was a fully connected layer that acted as a binary classifier. The features extracted by the first two layers (feature extractor) had a high abstraction level; thus, they could be reused for further diagnosis. Therefore, we froze the ADM weights after the training process. This tip, introduced in a previous work [17], helped reduce the overall computational cost in the long-term monitoring process owing to the dominance in the occurrence frequency of a normal state compared to an abnormal state.

### 3.3. Multi-Output Classification for Compound FD with Degradation Levels

The proposed multi-output classification model had two major stages: feature extraction and multi-output classification. The input of this model was the feature map extracted by the feature extractor of the ADM.

#### 3.3.1. Feature Extraction

The feature extractor of this model was established by stacking several groups of convolutional layers and max-pooling layers. Each group began with convolutional layers used to convolve across the input feature and result in a new feature map. Subsequently, max-polling supported down-splitting the created feature map by taking advantage of improving efficiency and controlling overfitting phenomena.

With input image data $A^{m-1}$ consisting of channels $k_m$, the output $A^m$ setting with the $o_m$ channel after a convolutional layer was calculated as

$$A_o{}^m = g_m \left( \sum_k W_{ok}{}^m * A_m{}^{m-1} + b_o{}^m \right) \tag{1}$$

where $g_m(.)$ denoted a nonlinear function, $W_{ok}{}^m$ was the weight matrix, $b_o{}^m \in \Re$.

Then, the output feature map $A^m$ was applied to the max-pooling operation for spatial dimensionality reduction. The max-pooling formula was

$$X^m = \max(A^m, s) \tag{2}$$

where $X^m$ denoted the output of the max-pooling operation and $s$ the pooling size of a non-overlapping segment. The operation $\max(.)$ returned the maximum value among the values of each non-overlapping segment.

#### 3.3.2. Multi-Output Classification

Conventionally, the classification problem was known using multiclass classification. It was a classification task with more than two classes, and each sample was labeled only by one class. Regarding the field of bearing fault type diagnosis, the classification task applied for a set of fault signal samples might either be of a specific type of fault (inner race fault, roller fault, outer race fault, etc.). Multi-class classification assumed that each fault signal sample belonged to only one label. With this definition, adopting multiclass classification for the diagnosis of both bearing fault type and degradation levels required training data for numerous classes. For example, with seven types of compound-bearing faults, three levels of degradation for each fault type, and one normal state, we needed to prepare data for classifying 22 separate classes [10]. In addition to the problem of preparing data for some classes, another adverse effect was that—in multiclass classification—each sample focused on representing fault features for individual classes without considering

the relationship with other aspects of the fault. Therefore, missing labels in some classes could cause serious problems for the overall prediction accuracy.

In contrast, multi-output classification was related to the ability of the estimator to handle several joint classification tasks. This tactic attempted to fit the input sample into one classifier per target by allowing multiple target variable classifications. The only predictor was trained to estimate a series of target functions (loss function 1, loss function 2, ... ) to predict a series of corresponding labels (output 1, output 2, ... ). Because of the use of the common predictor, correlated features were utilized to enhance the overall performance of all related targets. There was the fact that the probability of overfitting when using shared parameters between tasks was smaller than the risk of task-specific parameters [50]. Intuitively, the more tasks that were trained simultaneously, the greater the probability that the model found the representation capturing all related tasks.

After the feature extraction stage, a flattening operation was initially applied to create a 1D vector

$$z^m = flatten(X^m) \tag{3}$$

where $flatten(\cdot)$ denoted the flattening operation converting the matrix $X^m$ to a vector $z^m$.

The final layer of the architecture was divided into two sub-outputs that served $N$ targets. Thus, each input $X$ finally obtained the $N$ output as $y_{final\_output}^m = [y_1^m, y_2^m, \ldots, y_N^m]$. In the scope of this research, $N = 2$ for two targets of compound fault type classification and the level of degradation classification.

Concerning the training process, multi-output learned the way to match each input to multiple outputs. Given the input space $X \in \mathbb{R}^d$, and the output label space $Y \in \mathbb{R}^m$. The goal of the training process was to determine a function $f : X \to Y$ from the training set domain $D = \{(x_i, y_i) | 1 < i < n\}$, where $x_i \in X$ and $y_i \in Y$. Intuitively, the training problem became an optimizing function $F : X \times Y \to \mathbb{R}$ by using training samples. Subsequently, with unseen samples of validating or testing set $x$, the model was predicted $\widetilde{y} = f(x) = \text{argmax}_{y \in Y} F(x, y)$.

The overall loss function was defined as the sum of the sub-output loss of the training targets, $loss_{overall} = loss_{fault\_type} + loss_{crack\_size}$. For each sub-output, the loss function was obtained as

$$loss = -\frac{1}{M} \sum_{m=1}^{M} \sum_{p}^{P} 1\{y^m = p^m\} \cdot \log(\widetilde{y}^m) \tag{4}$$

where $M$ was the number of samples, and $P$ the set of labels in each sub-output, predicted value $\widetilde{y}$, true label $y$, and binary function $1\{\}$.

The process of minimizing the overall loss function was supported by an automatic stochastic line search (SLS) optimizer to automatically adjust the learning rate and enhance the convergence ability [51].

Concerning the SLS, stochastic gradient descent (SGD) is used to compute the gradient of the loss function of a minibatch in iteration $k$. It then updates the weights as $w_{k+1} = w_k - \eta_k \nabla f_{ik}(w_k)$, where $w_{k+1}$, and $w_k$ are the SGD iterates, $\eta_k$ is the step size, and $\nabla f_{ik}(\cdot)$ is the average loss function gradients computed at iteration $k$. Each stochastic gradient $\nabla f_{ik}(w)$ is assumed to be equivalent (e.g., $E_i[\nabla f_i(w)] = \nabla f(w)$ for all $w$).

The Armijo line [52] search is used as a criterion for deterministically setting the step size of gradient descent. At iteration $k$, the Armijo line search performs computations to choose a step size that satisfies the following condition, where $c > 0$ is a hyperparameter

$$f_{ik}(w_k - n_k \nabla f_{ik}(w_k)) \leq f_{ik}(w_k) - c \cdot \eta_k ||\nabla f_{ik}(w_k)||^2 \tag{5}$$

The adopted SLS optimizer can be summarized as: (1) compute the gradients $\nabla f_{ik}(w_k)$ for a given training batch; (2) search for a step size $\eta_k$ that satisfies the stochastic Armijo line search condition; and (3) use the step size and update the model parameters with SGD: $w_{k+1} = w_k - \eta_k \nabla f_{ik}(w_k)$.

## 4. Experiments and Results

In this section, we describe the data used to evaluate the proposed method. We show and discuss the results obtained from the experiments.

### 4.1. Experimental System and Dataset

Figure 3 shows that the data used for evaluating the proposed method were acquired from the experimental testbed. The experimental system consisted of a three-phase induction motor providing motion for the entire system, which changed the rotational speed by a controller. The motion from the motor was transferred into a gearbox (ratio 1.52:1) that contained two shafts; namely, drive-end and non-drive-end shafts. Each shaft was attached to two bearings (FAG NJ206-E-TVP2), and Table 1 lists the specifications. Moreover, the AE sensor was mounted near the bearing of the non-drive-end shaft. The non-drive-end shaft was connected to the load by a belt. The load was a blade that could be adjusted.

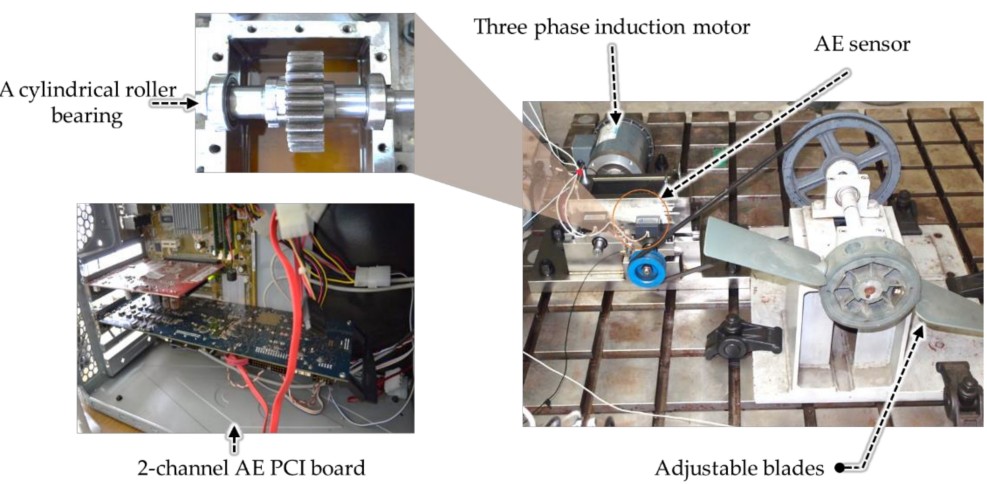

**Figure 3.** Data acquisition system and experimental testbed.

**Table 1.** Bearing specification.

| Contact angle | 0° |
|---|---|
| Number of rolling elements | 13 |
| Pitch diameter | 46.5 (mm) |
| Rolling element diameter | 9.0 (mm) |

Regarding the data acquisition system (DAS), a wideband AE sensor (PAC WSα) was used, with a frequency range of 100 kHz to 900 kHz, a peak sensitivity of −62 dB, a frequency response range of 1 kHz to 3 MHz, directionality of ±1.5 dB, and resonant frequency of 650 kHz. DAS was a PCI-2-based board with a sampling rate of 250 kHz.

Diamond cuts were used to create seven fault types (single and compound faults), and each fault type had three different crack sizes. Thus, there were 21 fault cases in total. We used a dataset for training, validating, and testing the multi-output CNN model (see Table 2). Concerning the training and validation subsets, the samples were acquired at rotational speeds of 300, 400, and 500 rpm; whereas, the testing subset samples were acquired at rotational speeds of 250, 350, and 450 rpm. The defined dataset helped evaluate the capability of the proposed method under the condition of variable rotational speeds. In the experiments, we used 80 samples for every fault case in the training process (60 samples of each fault case for training and 20 samples of each fault case for validation). After training, 400 samples of each fault case were used to test the trained model. Therefore, the training subset, validation subset, and testing subsets contain 1760, 440, and 8800 samples, respectively. Each sample was a spectrogram of a 0.05 s signal segment determined by the

range of the fault frequency components [21]. As Table 3 shows, 22 bearing states were introduced. The ADM was trained to classify the two states (normal and faulty states); whereas the multi-output model was trained to classify 21 fault states.

**Table 2.** Experimental compound fault with various levels of degradation dataset.

| Single and Compound Bearing Defect Dataset | Rotational Speed (rpm) | Crack Size | | |
|---|---|---|---|---|
| | | Length (mm) | Width (mm) | Depth (mm) |
| Training subset | 300, 400, 500 | 3 | 0.6 | 0.3 |
| | | 6 | 0.6 | 0.5 |
| | | 12 | 0.6 | 0.5 |
| Validation subset; Testing subset | 250, 350, 450 | 3 | 0.6 | 0.3 |
| | | 6 | 0.6 | 0.5 |
| | | 12 | 0.6 | 0.5 |

**Table 3.** Bearing fault dataset labels.

| Abbr. | Fault Category (Fault Type—Crack Size) | Multi-Output Labels | |
|---|---|---|---|
| | | Fault Type | Crack Size |
| BCO (3) | Outer raceway (3 mm) | 1 | 1 |
| BCI (3) | Inner raceway (3 mm) | 2 | 1 |
| BCR (3) | Roller (3 mm) | 3 | 1 |
| BCIO (3) | Inner and outer raceway (3 mm) | 4 | 1 |
| BCOR (3) | Outer raceway and roller (3 mm) | 5 | 1 |
| BCIR (3) | Inner raceway and roller (3 mm) | 6 | 1 |
| BCIOR (3) | Inner, outer raceway, roller (3 mm) | 7 | 1 |
| BCO (6) | Outer raceway (6 mm) | 1 | 2 |
| BCI (6) | Inner raceway (6 mm) | 2 | 2 |
| BCR (6) | Roller (6 mm) | 3 | 2 |
| BCIO (6) | Inner and outer raceway (6 mm) | 4 | 2 |
| BCOR (6) | Outer raceway and roller (6 mm) | 5 | 2 |
| BCIR (6) | Inner raceway and roller (6 mm) | 6 | 2 |
| BCIOR (6) | Inner, outer raceway, roller (6 mm) | 7 | 2 |
| BCO (12) | Outer raceway (12 mm) | 1 | 3 |
| BCI (12) | Inner raceway (12 mm) | 2 | 3 |
| BCR (12) | Roller (12 mm) | 3 | 3 |
| BCIO (12) | Inner and outer raceway (12 mm) | 4 | 3 |
| BCOR (12) | Outer raceway and roller (12 mm) | 5 | 3 |
| BCIR (12) | Inner raceway and roller (12 mm) | 6 | 3 |
| BCIOR (12) | Inner, outer raceway, roller (12 mm) | 7 | 3 |
| BNC | Normal state | | |
| BAC | Abnormal state | | |

*4.2. Evaluation Metrics*

To evaluate the proposed method, we considered the performance in each particular task: anomaly detection, fault type diagnosis, and degradation level diagnosis. Sensitivity was one of the most important metrics for diagnosis; therefore, it was used to evaluate and compare the diagnostic performance.

$$Sensitivity = \frac{N_{True\_Positive}}{N_{True\_Positive} + N_{False\_Negative}} \times 100(\%)$$ (6)

where $N_{True\_Positive}$ was the quantity of correctly predicted samples in a specific class, and $N_{False\_Negative}$ the quantity of incorrectly predicted samples in a specific class.

For the whole dataset, the average classification accuracy was measured as the average value of sensitivities.

$$ACA = \frac{\sum Sensitivity}{\sum N_{Classes}}$$ (7)

Moreover, concerning the multi-output model, we also evaluated the overall performance of the proposed model by defining a correct prediction as meaning that both labels of fault type and label of crack size were the same as their corresponding ground truth simultaneously. Therefore, the accuracy was calculated by

$$ACC_{multi\_output} = \frac{1}{M} \sum_{m=1}^{M} 1\left\{\widetilde{y}^m_{fault\_type} = p^m_{fault\_type}\right\} \times 1\left\{\widetilde{y}^m_{crack\_size} = p^m_{crack\_size}\right\}$$ (8)

*4.3. Classification Results by Using Proposed Multi-Output Classification*

We evaluated the performance of the proposed method based on the testing subset after the training process, with the experiments being repeated 10 times to obtain results on average. We initially evaluated the performance of each task (fault type diagnosis and degradation level diagnosis). The confusion matrices in Figure 4 show the results of these tasks. First, concerning the task of anomaly detection, owing to the simplicity of binary classification, absolute accuracy was easily achieved after several training epochs (five epochs). Second, owing to the correlated learning features, the task of fault type diagnosis also achieved 100% in all classes of single faults and compound faults. The task of the level of degradation diagnosis was shown to have greater difficulty in detecting some fault types in the early stage (small crack size). Nevertheless, it still achieved an accuracy of 97.62% for a 3 mm crack size and an overall accuracy of 99.21%.

In addition to evaluating diagnosis tasks separately, we provided an evaluation based on the combination of two tasks in the multi-output model to compare with conventional multi-class learning-based methods. This meant that there were 21 classes (seven fault types and three levels of gradation for each fault type) evaluated by the metrics of $ACC_{multi\_output}$. The confusion matrix in Figure 5 illustrates that the model had some missing predictions in cases of outer race faults or when all types of faults appeared. Figure 6 shows that the SVM-based method established by grid-search achieves an accuracy of 84.59% on average. The proposed multi-output CNN-based method also outperforms the previous multi-class CNN-based method [8] in solving the FD problem accompanied by the level of degradation. Our proposed multi-output CNN-based method can achieve an accuracy of diagnosis of up to 99.32%.

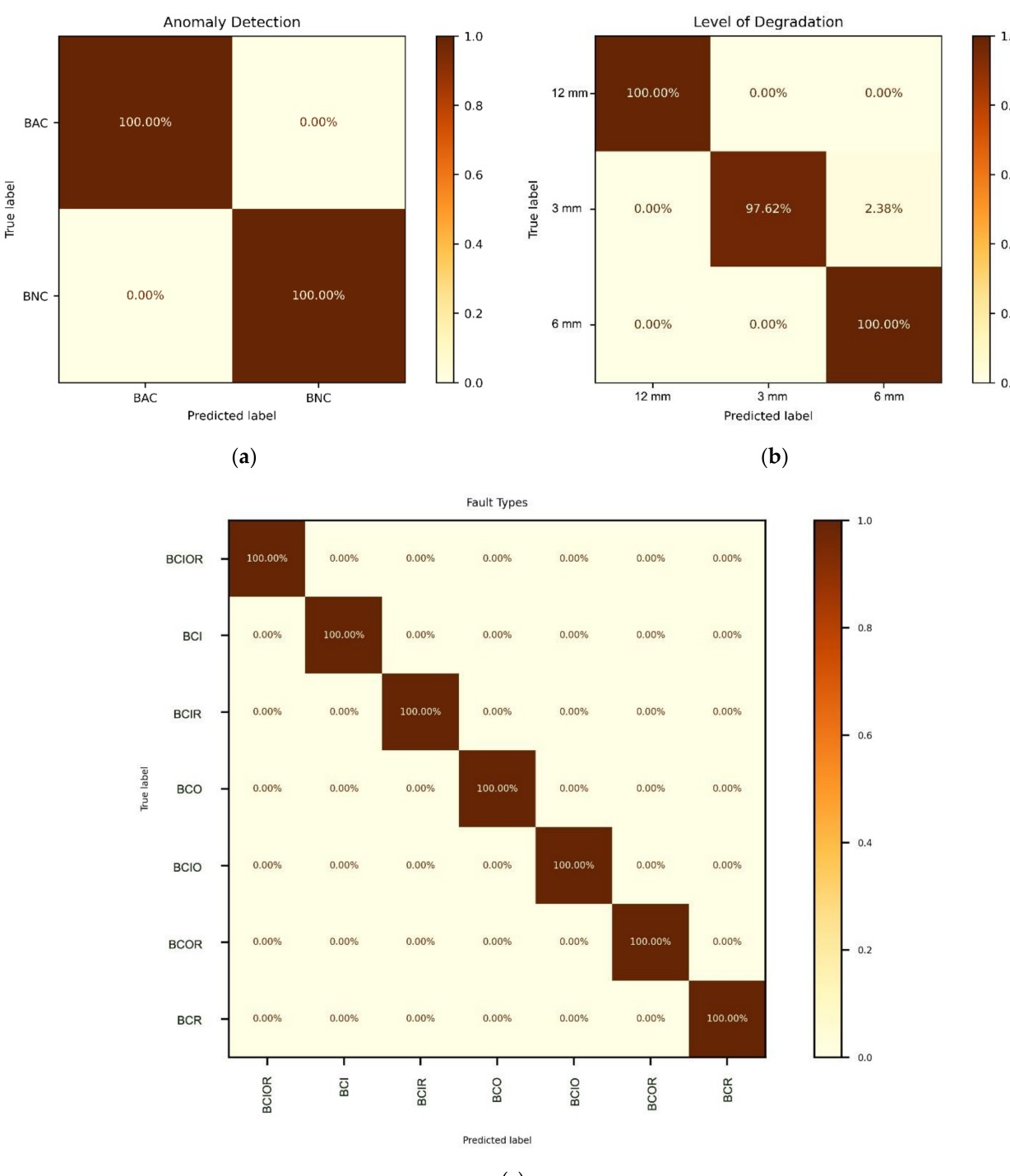

**Figure 4.** Confusion matrix showing the classification results for separate diagnosis tasks: (**a**) anomaly detection, (**b**) level of degradation classification, (**c**) compound fault type classification.

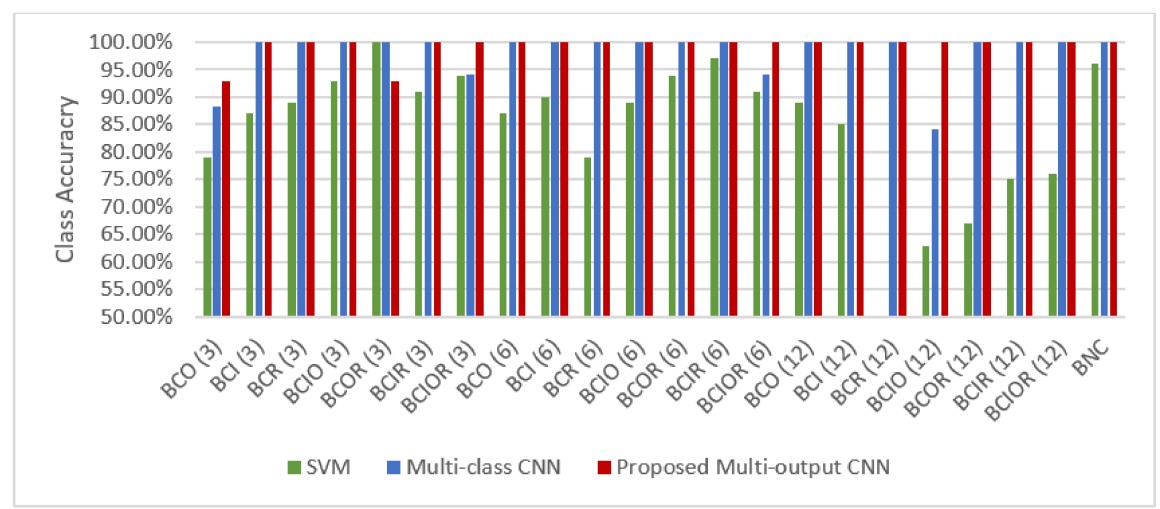

**Figure 5.** Confusion matrix showing the result of FD with the corresponding level of degradation simultaneously.

**Figure 6.** Diagnosis comparison for SVM-based method and two CNN-based methods.

The computational resource requirement was also an important aspect that needed to be considered when evaluating the applicability of an intelligent FD method. It was convenient to consider the number of MAC and the number of parameters of the two models separately (ADM and multi-output model). As Table 4 shows, the ADM using only one CNN layer accounted for approximately 0.05M MAC and 962 parameters. Not only MAC but also the number of parameters would affect inference latency owing to the dataflow latency, especially concerning memory bottleneck systems. If an upcoming input sample was predicted to be in the normal state, the inference process would be terminated by ADM. Otherwise, the features extracted by the feature extractor of ADM would be reused as input features of the following multi-output model, where the further inference process classified the fault type and level of degradation. In this way, the difference between the number of normal and abnormal states in reality (most of the time bearing was in a normal state) gave rise to saving computational resources [21]. However, when considering the tasks of FD and level of degradation diagnosis, the proposed multi-output CNN-based method also reduced the number of MAC (to 1.9M) and the number of parameters (to 2.31M) significantly compared to the conventional multi-class CNN (Efficient Net B0)-based method [10].

**Table 4.** Computational consumption comparison between CNN-based methods.

|  | Number of MAC | Number of Parameters |
| --- | --- | --- |
| Multi-class CNN [10] | 195M | 5.3M |
| Proposed multi-output CNN | 1.9M | 2.31M |
| Proposed ADM | 0.05M | 962 |

*4.4. Stability in Noisy Working Environments*

The proposed method's stability was also inspected in a noisy virtual environment to ensure that the model would be sustainable in real working environments in the presence of noise. We conducted the experiment by adding white Gaussian noise at various signal-to-noise ratios (SNR), which depended on the power of the meaningful signal and the power of noise

$$SNR_{dB} = 10 \log_{10} \left( \frac{P_{signal}}{P_{noise}} \right) \tag{9}$$

The results on average (see Table 5) proved that the accuracy of prediction experienced a marginal decrease for both types of models when the SNR was lower (i.e., the power of the meaningful signal was lower). However, the proposed multi-output outperformed the previous CNN-based method and still achieved an accuracy of 95.87%, while the SNR value was positive (i.e., the power of the signal was higher than the power of noise).

**Table 5.** Comparison of average accuracy between two CNN-based methods under noisy conditions.

| SNR | ACA | |
| --- | --- | --- |
|  | **Multi-Class CNN [10]** | **Proposed Multi-Output CNN** |
| No noise | 98.21 | 99.32 |
| 10 | 96.08 | 98.65 |
| 5 | 95.36 | 97.82 |
| 0 | 93.33 | 95.87 |

## 5. Conclusions

This paper proposed a multi-output CNN-based method for bearing fault multitask diagnosis under variable rotational speeds. The combination of using time–frequency representation and learning correlated features in multiple tasks of diagnosis helped improve the efficiency and performance of the proposed method. Pre-screening to classify between the normal state and anomaly also contributed to the efficiency of the entire

diagnosis process in the long run. The experiments' results indicated that the proposed bearing FD method was able to reduce the number of MAC and parameters compared to the conventional CNN-based method in the same tasks while maintaining higher accuracy in prediction (99.32% on average). The proposed method also proved its stability in noisy working environments. These improvements indicated that it was a reliable solution for smart factories in monitoring issues.

**Author Contributions:** Conceptualization, M.-T.P. and C.-H.K.; Formal analysis, J.-M.K. and C.-H.K.; Funding acquisition, C.-H.K.; Methodology, M.-T.P.; Software, M.-T.P.; Validation, C.-H.K.; Writing— original draft, M.-T.P.; Writing—review and editing, J.-M.K. and C.-H.K. All authors have read and agreed to the published version of the manuscript.

**Funding:** This research was supported by the National Research Foundation of Korea (NRF) grant funded by the Korean government (MSIT) (no. 2021R1A2C1009031).

**Institutional Review Board Statement:** Not applicable.

**Informed Consent Statement:** Not applicable.

**Data Availability Statement:** Not applicable.

**Conflicts of Interest:** The authors declare no conflict of interest.

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
