# Peer review of "2D CNN-Based Multi-Output Diagnosis for Compound Bearing Faults under Variable Rotational Speeds"

_machines, doi:10.3390/machines9090199_

Round 1

Reviewer 1 Report

Conclusively, I suggest this article may be accepted after a minor to moderate revision. After reviewing this article, I found the following revisions should be made:

  1. This article employs an automatic stochastic line search optimizer. It is better to add pseudocodes to describe the implementation of this optimizer.
  2. In Eq. (6), the authors define an ACA (averaged classification accuracy). However, after reviewing the entire article, I can't find this ACA has been used.
  3. Please add the size of the dataset employed in this manuscript.
  4. This article lacks some baseline models. I suggest these baseline models of different types (for example, the support vector machines model and random forest methods). Thus, the results in this manuscript can be highlighted.
  5. What does the "MAC" stand for? Please explain it.
  6. The authors compare the performance of a previous multi-class CNN and proposed multi-output CNN. I suggest the authors point the difference between these two types of CNN. Does this difference induce the difference between red and blue columns in Fig. 6?
  7. Please check the format of the reference list. This format seems to satisfy the standard of this journal.

Reviewer 2 Report

In a controlled experimental environment, the author converts signals from acoustic emission sensor near roller bearing to spectrograms which are input of 2D-CNN for bearing fault diagnosis. The results show7 fault types and 3 levels of degradation (total 21 cases) are accurately detected and classified. 

The paper could improved further if the following questions could be addressed in the paper. 

  1. 3mm, 6 mm and 12 mm length of crack size samples were used to train the model and the testing data set also are prepared with crack size exactly 3 mm, 6 mm and 12 mm. Curious if crack size of 4.5 mm, 9 mm are prepared, what would be result be? How would the model be used to address practical crack size scenario? 
  2. For one shaft, there are two bearings. Would you mind clarify if the crack is made on one of the bearing or both of the bearing? In figure 3, would you mind providing a close image of how the AE sensor is mounted? Is it close to one bearing than the other? What would the practical scenario be? 
  3. Could the author provide one or two sentence stating why the crack’s direction is chosen to   be in direction parallel to the shaft direction?
  4.  When reader first sees Figure 1, they have no idea what BCO, BCI, 3mm, 6mm, 12mm means, it would be better to reference later part of the paper in the caption of Figure 1.
  5.  The “Pulse”, “Accelerometer”, “Amplifier” in Figure 3 are not addressed in the caption of Figure 3. What is the purpose of the “adjustable blade”?
  6.  Possible to include software toolsets/environment used for the model?

Reviewer 3 Report

Dear Authors

I can congratulate the Authors for a very interesting work. Almost everything that is needed for this kind of research is presented in it. Thus the following was presented: 1)              Own proposals (e.g. CNN with multi-output classification and anomaly detection model) against the background of literature, perhaps too modest (but more on that later), concerning the issue discussed in the manuscript. The proposals announced in the introduction were implemented in 100%. 2)     Justification for the use of multi-output classification instead multi-class classification 3) Experimental testbed with the date aquisition system and a detailed description of the test. 4) Test results (bearing faults) in the table 5) Classification results by using proposed multi-output classification, confusion matrices and diagnosis comparison between two CNN-based methods on the graph  6) Comparation of average accuracy between two CNN-based methods at various signal-to-noise ratios. According to the reviewer, it would be advisable to replace the part of the literature more relevant to the topic of the paper (below [1 - 12]). In the opinion of the reviewer, it is impossible to conceal the Authors' own paper [2], which was the precursor to this one. Thanks to the publication of this literature [1 to 12], the manuscript will only gain, as the originality of the Authors' proposals will be more documented.   There is a small inaccuracy in verse 81 to correct, as it seems that it should have been written M.T. Pham et. al. [8] ....

Yours sincerely, Reviewer

1) Accurate Bearing Fault Diagnosis under Variable Shaft Speed using Convolutional Neural Networks and Vibration Spectrogram Minh Tuan Pham, Jong-Myon Kim and Cheol Hong Kim Appl. Sci. 202010(18), 6385;

2) Convolutional Neural Networks for Automated Rolling Bearing Diagnostics in Induction Motors Based on Electromagnetic Signals. Marcello Minervini,Maria Evelina Mognaschi,Paolo Di Barba and Lucia Frosini Sci.202111(17), 7878;

3) Bearing Fault Diagnosis Based on Improved Convolutional Deep Belief Network Shuangjie Liu, Jiaqi Xie, Changqing Shen, Xiaofeng Shang, Dong Wang and Zhongkui Zhu Appl. Sci. 202010(18), 6359;

4) Bearing Fault Diagnosis Using Grad-CAM and Acoustic Emission Signals JaeYoung Kim and Jong-Myon Kim Appl. Sci. 202010(6), 2050;

5) A Novel Fault Diagnosis Algorithm for Rolling Bearings Based on One-Dimensional Convolutional Neural Network and INPSO-SVM Yang Shao, Xianfeng Yuan, Chengjin Zhang, Yong Song and Qingyang Xu Appl. Sci. 202010(12), 4303;

6) An Improved Scheme for Vibration-Based Rolling Bearing Fault Diagnosis Using Feature Integration and AdaBoost Tree-Based Ensemble Classifier Bingxi Zhao, Qi Yuan and Hongbin Zhang Appl. Sci. 202010(5), 1802;

7) Rolling Bearing Fault Diagnosis Based on Wavelet Packet Transform and Convolutional Neural Network Guoqiang Li, Chao Deng, Jun Wu, Zuoyi Chen and Xuebing Xu Appl. Sci. 202010(3), 770;

8) Fault Diagnosis of Induction Motor Using Convolutional Neural Network Jong-Hyun Lee, Jae-Hyung Pack and In-Soo Lee Appl. Sci. 20199(15), 950;
9) Application of Multi-Dimension Input Convolutional Neural Network in Fault Diagnosis of Rolling Bearings Tao Zan, Hui Wang, Min Wang, Zhihao Liu and Xiangsheng Gao Appl. Sci. 20199(13), 2690;

10) A Deep Learning Method for Bearing Fault Diagnosis through Stacked Residual Dilated Convolutions Zilong Zhuang, Huichun Lv, Jie Xu, Zizhao Huang and Wei Qin Appl. Sci. 20199(9), 1823;

11) A Fault Diagnosis Approach for Rolling Bearing Based on Convolutional Neural Network and Nuisance Attribute Projection under Various Speed Conditions Huijie Ma, Shunming Li and Zenghui An Appl. Sci. 20199(8), 1603;

12) Bearing Fault Diagnosis under Variable Rotational Speeds Using Stockwell Transform-Based Vibration Imaging and Transfer Learning Md Junayed Hasan and Jong-Myon Kim Appl. Sci. 20188(12), 2357.
